# Gender Differences in Quality of Life and Health Services Utilization among Elderly People in Rural Vietnam

**DOI:** 10.3390/ijerph16010069

**Published:** 2018-12-28

**Authors:** Thang Pham, Nguyen Thao Thi Nguyen, Sophie Bao ChieuTo, Tuan Le Pham, Thanh Xuan Nguyen, Huong Thu Thi Nguyen, Tam Ngoc Nguyen, Thu Hoai Thi Nguyen, Quang Nhat Nguyen, Bach Xuan Tran, Long Hoang Nguyen, Giang Hai Ha, Carl A. Latkin, Cyrus S.H. Ho, Roger C.M. Ho, Anh Trung Nguyen, Huyen Thi Thanh Vu

**Affiliations:** 1Department of Gerontology and Geriatrics, Hanoi Medical University, Hanoi 100000, Vietnam; phamthang@hmu.edu.vn (T.P.); xuanthanh1901vlk@gmail.com (T.X.N.); nguyenthuhuong.hmu@gmail.com (H.T.T.N.); ngoctamyhn@gmail.com (T.N.N.); nththu.bvlk@gmail.com (T.H.T.N.); 2National Geriatric Hospital, Hanoi 100000, Vietnam; 3Duke University, Durham, NC 27708, USA; ttn5@duke.edu; 4Department of Social and Behavioral Sciences, Yale School of Public Health, New Haven, CT 06510, USA; sophie.to@yale.edu; 5Department of Family Medicine, Hanoi Medical University, Hanoi 100000, Vietnam; phamletuan@moh.gov.vn; 6Dinh Tien Hoang Institute of Medicine, Hanoi 100000, Vietnam; 7Université Claude Bernard Lyon 1, 69100 Villeurbanne, France; quang.n.nguyen@alumni.duke.edu; 8Institute for Global Health Innovations, Duy Tan University, Da Nang 550000, Vietnam; giang.ighi@gmail.com; 9Institute for Preventive Medicine and Public Health, Hanoi Medical University, Hanoi 100000, Vietnam, 100000; bach.ipmph@gmail.com; 10Bloomberg School of Public Health, Johns Hopkins University, Baltimore, MD 21205, USA; carl.latkin@jhu.edu; 11Center of Excellence in Behavioral Medicine, Nguyen Tat Thanh University, Ho Chi Minh City 770000, Vietnam; longnh.ph@gmail.com (L.H.N.); hocmroger@yahoo.com.sg (R.C.M.H.); 12Department of Psychological Medicine, National University Hospital, Singapore 119074, Singapore; cyrushosh@gmail.com; 13Department of Psychological Medicine, Yong Loo Lin School of Medicine, National University of Singapore, Singapore 119228, Singapore

**Keywords:** aging, elderly, gender differences, rural, quality of life, health services utilization, Vietnam

## Abstract

As in much of the world, the elderly population in Vietnam is growing rapidly with two-thirds of them currently living in rural areas. Besides limited access to quality healthcare services, they also have unique health profiles and needs due to various factors, including the highly skewed sex ratio of more women residing in rural areas. However, the relationship between gender, health-seeking behaviors, and health outcomes in this under-served population has not been well characterized. This study sought to explore the associations of gender with health-related quality of life and health-seeking behavior among the elderly in Soc Son, a rural district of Hanoi, Vietnam. A cross-sectional design was used; elderly individuals were surveyed across the domains of socioeconomic information, health status, and healthcare service utilization. We found that overall, women had poorer health and quality of life even though gender difference did not appear to significantly influence their levels of health services utilization. A greater understanding of the systemic, sociocultural, and psychological factors underlying such differences may help better inform future healthcare service delivery strategies targeting this growing population in rural areas.

## 1. Introduction

The global elderly population is growing rapidly, with the world gaining one million older persons each month [1]. National governments, healthcare systems, health professionals, and families are tasked with the challenge of providing support for this group, who have distinct characteristics such as increased likelihood of having a chronic condition [1]. Vietnam is no exception to these trends. Between 1979 and 2015, the country’s aging index (number of people aged 60+ per hundred children aged under 15) has almost tripled—from 17 to 47—and is projected to increase to 138 by 2049 [2]. And the aged dependency ratio (ratio of non-working age individuals to working age), which was previously stable between 1979 and 2009 at approximately one person aged 60+ per 10 working-age people, has recently decreased to 1:9 with projections that it will drop to 1:3.5 by 2049 [2]. As a result, it has become increasingly difficult for the national government to adequately fund public services for the elderly [2]. Currently, all individuals aged 80 and older qualify for government-sponsored health insurance, but among those 60–79 years old, only poor individuals without family support and individuals with disabilities receive assistance [2].

The Ministry of Health (MOH) has taken steps to better serve the elderly population—for example, requiring hospitals and clinics to designate a certain number of beds/rooms for elderly patients [2]. By 2020, all provincial and tertiary hospitals should have established geriatric departments [2]. However, at present, the MOH has yet to establish standards of care for these departments. Further, while three universities in Vietnam do provide training in geriatric care, there are currently no competency standards and no standardized curriculum for the medical specialty [2].

In rural areas, where two-thirds of the elderly population reside, these health challenges are particularly compounded by the lower availability of resources [2]. While urban areas are home to only about one-third of the population, 53% of physicians are concentrated in those areas [2]. Policies have been implemented to incentivize physicians to relocate to rural communities, including higher salary and mandatory rural service, but the uneven distribution remains [3]. Despite developments in communication and expansion of healthcare education—such as implementation of telemedicine and training of community health workers—rural residence has been found in recent studies to be a predictor of worse health [4].

Further complicating the health landscape among the elderly population is this group’s skewed sex ratio, which is more pronounced in rural communities than in urban communities among those 65 and older: As of 2015, among those aged 60–64, the sex ratio (men per 100 women) is 85 in rural areas versus 83 in urban areas; but for all older age groups, the sex ratio becomes more skewed among those living in rural areas [5]. For example, among those 85 and older, the sex ratio is48 in rural areas and 53 in urban areas [5]. The skewed sex ratio may be partially explained by the Vietnam War, given that the current elderly generation was in their 20s to 30s at the time. Due to the civil war and later conflicts in Cambodia and China, there was a greater percentage of male mortality in that generation [6]. It has been estimated that from 1955–2002 across 13 countries, including Vietnam, 58% of violent war deaths occurred in those 15-34 years old, and 81% of violent war deaths occurred in men [7]. Men were also more likely to emigrate from Vietnam during and after the war to avoid being drafted and due to greater ease of mobility [6]. As a result, the sex ratio after the war decreased, as evident when using neighboring Thailand as a comparison [6]. Women’s higher life expectancy, which is a global phenomenon, may also partially account for the skewed sex ratio. In particular, women in Vietnam have a life expectancy at birth of 81 years versus 72 years for men as of 2016, per the World Bank [8,9]. Around the world, women have higher life expectancy than men, but they are susceptible to different disease patterns and may practice different health-seeking behaviors than men [10]. For example, women are more likely to have arthritis and osteoporosis [10]. In addition, previous work has suggested that in response to tuberculosis infections, Vietnamese women tend to initially seek self-medication or health services, whereas men tend to neglect symptoms until they progress to serious disease [11]. One Vietnam-specific historical phenomenon potentially relevant to differential health status and health-seeking behaviors between men and women is that while men were more likely to suffer directly from the effects of the Vietnam War—both psychological and physical injury—women were also indirectly affected, often becoming the primary caregiver for their family due to the injury or death of their family members [6]. Thus, while veterans also face bureaucratic obstacles in obtaining healthcare, their injuries are often more apparent, whereas for women, the stress and PTSD associated with caring for family members might pass undetected [6]. However, the relationship between gender, health-seeking behavior, and health outcomes between men and women remains unclear.

In this study, we aimed to explore the associations of gender with health-related quality of life, including current health status and history of illnesses, and health-seeking behaviors among the elderly in a rural district in Vietnam. Such knowledge might help inform future healthcare services and policies to better address the unique growing needs of elderly individuals, especially those living in rural areas.

## 2. Materials and Methods

### 2.1. Study Design

This cross-sectional study was conducted in Soc Son, Ha Noi, Vietnam, from February to April 2017. The study included a total of 523 elderly individuals over 60 years old who were living in Soc Son district, Hanoi, Vietnam. Soc Son district is a typical rural setting in the northern Vietnam. Participants with unstable medical conditions, severe chronic diseases (e.g., coronary heart disease, cardiomyopathy, recent stroke) and cognitive impairment were excluded from participation in this study. These conditions might significantly affect the ability to understand and answer he questionnaire. First, all elderly people residing in Soc Son district who met the inclusion criteria were listed (around thirty-three thousand people). Then, by using a computer software, we selected randomly 530 elderly people and went to their house with the support of local health workers for interviews. Finally, a total of 523 (98.7%) elderly individuals agreed to participate in the study (30.0% men, 70% women).

### 2.2. Measures and Instruments

Face-to-face interviews using a structured questionnaire were conducted, which lasted 25 min per interview, to collect data about current socioeconomic information, health status, and healthcare service utilization. Data collectors, who were trained to ask questions consistently, included undergraduate medical and nursing students at the Hanoi Medical University. We did not involve physicians and nurses in the hospital to collect data to prevent potential social desirability bias.

*Socioeconomic and behavior information*: Sociodemographic and behavior characteristics of the participants included age, gender, current occupation, religion, marital status, whether they had a caregiver and who that person was, poverty status, and whether they had health insurance. We also measured cigarette smoking and alcohol consumption in the past five years.

*Health-related quality of life*: We employed the EuroQol-5 dimensions-5 Levels (EQ-5D-5L) to measure health-related quality of life (HRQOL) in five aspects: (1) Mobility; (2) Self-care; (3) Usual activities; (4) Pain/discomfort and (5) Anxiety/depression. Each of domain has five answer options from “No problem” (code 1) to “Extreme problems” (code 5) [12], and combining answers of these five domains could produce 3,125 possible health states from 11111 (full health) to 55555 (worst health) [12]. Each health state can be used to define one single “utility” score using Thailand cross-walk value set (score ranged from −0.451 (for 55555) to 1 (for 11111) [12]). Moreover, we categorized those answering “No problem” to the “No problem” group, and others to the “Having problems” group. The EQ-5D-5L instrument has been translated and validated in Vietnamese general population [13]. We also used EQ-visual analogue scale (EQ-VAS) which had a score ranging from 0 (the worse health state that patients could imagine) to 100 (the best health state that patients could imagine). According to Bach et al. (2015), employing both measures (EQ-5D-5L—an indirect preference instrument and EQ-VAS—a direct preference instrument) is recommended to evaluate the short-term and long-term changes of HRQOL [14]. In addition, we asked participants to report their history of illness.

*Healthcare service utilization*: We asked participants to report their healthcare service usage included any in-patient care (binary variable—Yes/No), out-patient visits (binary variable—Yes/No), and/or regular health examinations (binary variable—Yes/No), as well as number of each type of service use (continuous variable) twelve months prior to the interview. In addition, we asked about the medical facility that they first reported health problems, the quality of health services they received, their level of satisfaction with medical staffs, and the ease of visiting health facilities or receiving medical results at these facilities.

### 2.3. Data Analysis

STATA 12.0 software (Stata Corp. LP, College Station, TX, USA) was used for data analysis. Descriptive statistics consisting of frequency, percentage, mean, and standard deviation were performed. We compared the differences of health status, HRQOL and health service utilization between males and females using Chi-squared test and T-test. As the EQ-5D index and EQ-VAS scores data were censored, we employed the multivariable Tobit regression (or Censored regression) model in order to provide a better estimation than that from a typical linear regression when determining associated factors. Meanwhile, to determine factors associated with inpatient and outpatient service utilization in the last 12 months, multivariable logistic regression was used. These regression models were conducted in combination with stepwise forward selection strategies to reduce the models’ *p*-value for the log-likelihood ratio test that was less than 0.2 was used as a threshold for variable selection. Statistical significance was identified when the *p*-value was less than 0.05.

### 2.4. Ethical Approval

Each participant was assigned an identity number, and we removed all data about name, phone number or address in the raw dataset that can be used to identify the participants. The Institutional Review Board of National Geriatric Hospital approved this study protocol ((Reference: No 35; 16/01/2017). All patients were asked to give their written informed consents prior to participating in this study, acknowledging their understanding of the study and the ability to withdraw from the study at any time. All data were coded and secured in the study inventory at the National Geriatric Hospital from which only the principal investigator had access in order to protect the participants’ confidentiality.

## 3. Results

Table 1 shows that most of respondents were women (70.0%), farmers (43.2%) or retired (42.5%), and living with a spouse/partner (63.2%). Most participants had a caregiver (96.2%) and health insurance (84.3%). The mean age of participants was 71.0 (SD = 8.2) years old with men averaging 73.2 years and women 71.0 years. There were significant differences in occupation, marital status, having a caregiver, and type of caregiver between men and women (*p* < 0.01). Notably, all men had a caregiver in contrast to 94.5% of women. There were also significant differences between access to health insurance between men (93%) and women (80.6%) (*p* < 0.01). There was no significant difference in economic status between men and women.

As presented in Table 2, 5.2% of participants identified as current smokers and 17.8% alcohol drinkers. The proportions of having difficulty in mobility, self-care, usual activity, pain/discomfort and anxiety/depression were 50.3%, 24.7%, 44.9%, 72.3%, and 37.1%, respectively. Arthritis and hypertension were two most common illnesses in our respondents (28.7% and 32.3%, respectively). The mean EQ-5D index was 0.66 (SD = 0.22) and EQ-VAS was 61.2 (SD = 21.2). 

Significant differences in nearly all HRQOL indicators—history of stroke, arthritis and hypertension, smoking and alcohol use, and EQ-5D index—were found between men and women. Among HRQOL indicators, the following were found to be significantly different between men and women: difficulty with mobility (53.6% vs. 42.7%), difficulty with usual activities (48.4% vs. 36.9%), pain/discomfort (74.9% vs. 66.2%), and anxiety/depression (41.5% vs. 26.8%) (*p* < 0.05). The rates of stroke, hypertension, and arthritis were significantly different among men and women—0.3% of women had a history of stroke versus 5.7% of men, 31.1% of women had history of arthritis versus 22.3% of men, and 27.3% of women had a history of hypertension versus 44.0% of men. In terms of smoking and drinking status, 99.5% of women identified as non-smokers and non-drinkers in the last five years in contrast to 69.4% and 35.7% of men, respectively. Men also had an EQ-5D index significantly different from that of women (0.70 vs. 0.64).

Table 3 indicates that 49.5% of participants used outpatient services and 15.3% used inpatient services in the last 12 months. The mean number of inpatient, outpatient and regular checkup services used were 0.3 (SD = 1.1), 1.6 (SD = 2.7) and 3.6 (SD = 4.4), respectively. Of those who used health services, 5.6% and 4.6% were not satisfied with the quality of health services and medical staff, respectively, and 18.9% and 54.1% experienced difficulty in visiting health facilities and obtaining examination results at health facilities, respectively. There were significant differences in the first facility where health services were sought, satisfaction of health services, and satisfaction with medical staff between men and women (*p* < 0.05). There was no significant difference between difficulty visiting health facilities and receiving results among men and women.

Table 4 shows that men and former smokers had higher EQ-5D index in comparison to women and non-smokers. Meanwhile, higher age, self-employment status, having chronic lung diseases and arthritis, and history of drinking were negatively associated with EQ-5D index. Higher age, religion, having chronic lung diseases, arthritis, and smoking history were associated with lower EQ-VAS. No association between gender and EQ-VAS was found.

As shown in Table 5, no association was found between gender and health service use. People with higher HRQOL were less likely to use inpatient services in the last 12 months, while farmers and those with health insurance were more likely to use outpatient services. Self-employed, near-poor or non-poor people were less likely to have regular health check-ups. Those with health insurance were more likely to have regular health check-ups in the last 12 months.

## 4. Discussion

This study identified differences in sociodemographic characteristics, quality of life, health outcomes, and patterns of health services utilization between elderly women and men in the rural district of Soc Son. Future research should expand upon these findings and, moreover, characterize their underlying systemic, sociocultural, and psychological mechanisms, in order to develop solutions to improve the health of elderly persons in rural parts of Vietnam. Below, we discuss the findings of this study by grouping them into sections that give rise to similar implications for future research directions, policies, and interventions.

### 4.1. Social Support

Men and women had differential marital status; 85.4% of men surveyed were living with their spouse/partner in comparison to 53.7% of women. The differing proportions of elderly men and women living with a partner or spouse likely reflects Vietnam’s skewed sex ratio in this age group, as discussed earlier. Previous literature has long demonstrated a salient association between social support and positive health outcomes [15]. Thus, one possibility for the overall better health outcomes among men in the present study is greater social support, particularly in the form of spousal/partner support. A number of studies have examined in particular the relationship between social support and depression among elderly individuals, and have found that increased social support is associated with decreased depression [16]. Indeed, our study seems consistent with this evidence; the largest difference in health outcomes between men and women was observed in the anxiety/depression measure, with 41.5% of women and 26.8% of men reporting anxiety/depression (*p* < 0.01). However, we cannot draw any conclusions from this study regarding the specific relationship between social support and mental health outcomes in this group; deeper examination into the forces underlying each of these measures is warranted. 

Another related characteristic is caregiving. Though not as pronounced as the difference in marital status, the difference between women and men in having a caregiver (all men had caregivers in contrast to 94.5% of women) and in type of caregiver may also contribute to differential health outcomes. Caregiving needs to be measured in more detail in the future before any inferences can be made regarding its role in health outcomes in this population. It may be useful in future research to measure caregiver status, i.e., whether or not the participants are caregivers themselves, as this too may play a role in shaping health outcomes [17].

### 4.2. Differential HRQOL profiles

Women fared significantly (*p* < 0.05) worse across almost all HRQOL indicators. Here, the largest difference was observed in the anxiety/depression measure; as mentioned above, 41.5% of women and 26.8% of men reported anxiety/depression (*p* < 0.01). Furthermore, men had higher EQ-5D index scores in comparison to women. This points to a need for further characterization of women’s and men’s well-being and mental health, in order that the origins of such disparities may be elucidated so that the overall lower quality of life that women are experiencing can be addressed. However, this may also point to reporting bias; men with anxiety or depression may be less likely to self-report these than women with anxiety or depression.

In terms of chronic disease patterns, our study findings are consistent with trends previously observed: Arthritis was found to affect more female participants than male participants (though most previous studies have been conducted on Western populations) [18]. Hypertension and stroke were more prevalent among men, as seen in both studies conducted in Vietnam and throughout Asia [19,20,21,22]. As these findings corroborate disease patterns observed elsewhere, they may be able to be addressed by policies and interventions similar to existing ones that have been demonstrated effectiveness.

Furthermore, it appears necessaryto investigate the differential occupational hazards (or benefits) that elderly women and men living in rural areas may face, and the impact that such hazards could have on health: This study showed that occupation differed significantly by gender; of note, most men surveyed were retired, whilemost elderly women surveyed worked as farmers. The type and extent of work that elderly women and men do may affect both physical and psychological health.

This study also found a few rather counterintuitive results: First, while women have significantly lower EQ-5D scores than men, there was no significant difference in EQ-VAS scores between men and women. As discussed in a study of individuals with HIV/AIDS, this discrepancy may be due to the fact that because EQ-VAS is a direct preference-based measure, which directly asks participants to evaluate their own health statuses, it reflects participants’ perception of their health as opposed to their “objective” health status [23]. In contrast, EQ-5D uses an indirect approach by evaluating health utility, using preferences indicated by participants to predict health states. While direct and indirect measures should be used to more completely depict health status, EQ-5D likely more objectively depicts health state in this case. Thus, the lack of difference in EQ-VAS scores between men and women may suggest that women are less likely to perceive their health conditions to be as serious as they in reality may be. Second, even though more men than women were current or recent smokers and current or recent alcohol users, both men and former smokers had higher EQ-5D scores than women and non-smokers, respectively. This is after adjusting for covariates, as a multivariate regression model was used. These associations should be explored further for any possible confounding factors or other explanations.

In order to deliver care specific to the distinct needs of women and men and to better serve the rural elderly population as a whole, the results of this study should be a starting point for building a more comprehensive body of knowledge that can ultimately be incorporated into the training of providers who work with this population, the development of health promotion and disease prevention/management programs, and the creation of effective and appropriate policies.

### 4.3. Health Services Access and Utilization

The data indicates that while 65.6% of men are retired, only half this number—32.5%—of women have that same occupational status; 52.5% of women work as farmers. In addition, women are less likely to have health insurance than men. The fact that that men tend to work more white or blue-collar jobs whereas women tend to work as farmers likely accounts for this difference, as white or blue-collar jobs may be more likely to provide retirement pensions and healthcare coverage, leading to higher rates of retirement and insurance coverage among men.

Regarding healthcare utilization, women were more likely than men to report satisfaction (“satisfied” or “very satisfied”) with health services and medical staff. Additionally, there was no difference in healthcare utilization between men and women, which is particularly noteworthy because women have less access to caregivers, worse current health statuses, and worse EQ-5D scores—which point to the need to seek more health services—yet their amount of healthcare utilization is not reflective of this. In fact, women are more likely than men to try self-medication initially for health problems as opposed to seeking care at a health facility. Two findings from this study may point to why this may be the case: First, women are less likely to have health insurance, as discussed above; this lack of coverage may play a role in their lower utilization of health services, which could in turn negatively affect their health outcomes. Second, there is no difference in EQ-VAS scores between men and women, which may suggest that women are less likely to perceive their health conditions to be as serious as they in reality may be, which may in turn play a role in women’s lower healthcare utilization in relation to their poorer health status. Overall, the reasons for this underutilization need to be illuminated; future research could employ qualitative methods, such as conducting semi-structured interviews and focus groups with women, to better understand how and to which extent they are using healthcare services. Lastly, demographic characteristics, such as educational background, may also play a role in health-seeking behavior and health-related knowledge, and subsequently health outcomes.

Limitations of the present study include its cross-sectional nature, which does not allow us to illuminate any causal relationships. In addition, as only a handful of demographics, health behaviors, and outcomes were addressed, an accurate and comprehensive profile of this population’s health needs remains to be established. Moreover, the proportion of women participating in the study outweighed the number of men, which might lead to selection bias as well as under- or over- estimate the HRQOL and health utilization in both genders. A lack of information about education level in this study was also our limitation since this variable might affect our outcomes of interests. Lastly, this study was carried out in one district in northern Vietnam and on participants who in general were healthy and had high access to health insurance; when considering large-scale initiatives, policies, and interventions, it will be necessary explore healthcare utilization across other rural regions of Vietnam as well.

## 5. Conclusions

This study points to several important implications in the care of elderly patients. First, efforts should be made to ensure that elderly individuals lacking family support, disproportionately women, have access to caregivers and financial support. Consideration should be given to providing financial support especially to women, who are less likely to have jobs that provide benefits such as healthcare and retirement. Such initiatives should make sure to empower elderly persons (again, particularly women) in sustainable ways. Second, given the rapidly growing rural elderly population, there should be greater emphasis placed on training health staff in geriatric care in those areas. Moreover, given that most of the rural elderly population are women, healthcare professionals should also be trained specifically in health issues with greater prevalence among this subpopulation—such as arthritis, osteoporosis, and other symptoms related to menopause. Occupational health must also not be overlooked—women and men may have distinct health needs that arise from the type of labor they do (e.g., most men are retired, whereas most women work as farmers). Furthermore, while this study describes the underutilization of health services among female elderly patients with respect to their current health conditions, questions of why this pattern exists still remains. This study found that lack of health insurance is likely one barrier to care, but future research should work to explore other factors—structural, sociocultural, and psychosocial—that may also contribute to this trend.

## Figures and Tables

**Table 1 ijerph-16-00069-t001:** Sociodemographic characteristics of participants.

Characteristics	Women		Men		Total		*p*-Value
	*n*	%	*n*	%	*n*	%	
**Total**	366	70.0	157	30.0	523	100.0	
**Religion**							
No	316	86.3	140	89.2	456	87.2	0.37
Yes	50	13.7	17	10.8	67	12.8	
**Occupation**							<0.01
Retired	119	32.5	103	65.6	222	42.5	
Self-employed	19	5.2	12	7.6	31	5.9	
Farmers	192	52.5	34	21.7	226	43.2	
Others	36	9.8	8	5.1	44	8.4	
**Marital status**							<0.01
Single	23	6.3	3	1.9	26	5.0	
Living with spouse/partner	196	53.7	134	85.4	330	63.2	
Divorced/separated/widowed	146	40.0	20	12.7	166	31.8	
**Having a caregiver**							<0.01
No	20	5.5	0	0.0	20	3.8	
Yes	346	94.5	157	100.0	503	96.2	
**Family caregiver**							<0.01
Spouse and children	305	88.2	152	96.8	457	90.9	
Grandchildren	32	9.3	5	3.2	37	7.4	
Others	9	2.6	0	0.0	9	1.8	
**Economic status** *							0.99
Poor	34	10.4	14	10.1	48	10.3	
Near poor	36	11.0	15	10.9	51	10.9	
Non-poor	258	78.6	109	79.0	367	78.8	
**Active health insurance**							<0.01
No	71	19.4	11	7.0	82	15.7	
Yes	295	80.6	146	93.0	441	84.3	
	**Mean**	**SD**	**Mean**	**SD**	**Mean**	**SD**	
**Age**	70.1	8.0	73.2	8.3	71.0	8.2	<0.01

* Based on the Government’s classification.

**Table 2 ijerph-16-00069-t002:** Current health status and HRQOL of participants.

Characteristics	Women		Men		Total		*p*-Value
	*n*	%	*n*	%	*n*	%	
**Having problems in**							
Mobility	196	53.6	67	42.7	263	50.3	0.02
Self-care	99	27.1	30	19.1	129	24.7	0.05
Usual activities	177	48.4	58	36.9	235	44.9	0.02
Pain/Discomfort	274	74.9	104	66.2	378	72.3	0.04
Anxiety/Depression	152	41.5	42	26.8	194	37.1	<0.01
**History of illness**							
Stroke	1	0.3	9	5.7	10	1.9	<0.01
Anemic cerebral ischemia	23	6.3	5	3.2	28	5.4	0.15
Diabetes	13	3.6	11	7.0	24	4.6	0.08
Chronic lung disease	20	5.5	15	9.6	35	6.7	0.09
Parkinson’s	2	0.6	0	0.0	2	0.4	0.35
Arthritis	115	31.1	35	22.3	150	28.7	0.03
Hypertension	100	27.3	69	44.0	169	32.3	<0.01
**Smoking cigarettes in the last 5 years**							<0.01
Non-smoker in last 5 years	364	99.5	109	69.4	473	90.4	
Former smoker in last 5 years	0	0.0	23	14.7	23	4.4	
Current smoker	2	0.6	25	15.9	27	5.2	
**Drinking alcohol in the last 5 years**							<0.01
Non-drinker	360	98.4	56	35.7	416	79.5	
Former alcohol drinker	0	0.0	14	8.9	14	2.7	
Current alcohol drinker	6	1.6	87	55.4	93	17.8	
	**Mean**	**SD**	**Mean**	**SD**	**Mean**	**SD**	***p*-value**
**EQ-5D-5L-Index** ^a^	0.64	0.21	0.70	0.23	0.66	0.22	<0.01
**VAS Score** ^b^	61.1	21.0	61.6	21.6	61.2	21.2	0.81

^a^ Reported EQ-5D-5L-Index indicates the number of participants who reported having pain/discomfort and anxiety/depression. ^b^ EQ-VAS score ranges from 0 (the worse health state) to 100 (the best health state).

**Table 3 ijerph-16-00069-t003:** Health service utilization.

Characteristics	Women		Men		Total		*p*-Value
	*n*	%	*n*	%	*n*	%	
**Health service use in the past 12 months**							
Inpatient	53	14.5	27	17.2	80	15.3	0.43
Outpatient	191	52.2	68	43.3	259	49.5	0.06
Regular health check-up	254	69.4	120	76.4	374	71.5	0.10
**First medical facility where participants sought services for health problems**							0.03
The central hospital	7	1.9	2	1.3	9	1.7	
The provincial hospital	2	0.6	3	1.9	5	1.0	
District health department	67	18.3	49	31.2	116	22.2	
Commune health station	208	56.8	79	50.3	287	54.9	
Private health facility	20	5.5	7	4.5	27	5.2	
Self-medication	38	10.4	10	6.4	48	9.2	
Others	24	6.6	7	4.5	31	5.9	
**Satisfaction of health services**							0.02
Very unsatisfied	1	0.3	2	1.3	3	0.6	
Unsatisfied	0	0.0	1	0.7	1	0.2	
Neutral	11	3.2	13	8.7	24	4.8	
Satisfied	78	22.4	27	18.0	105	21.1	
Very satisfied	258	74.1	107	71.3	365	73.3	
**Level of satisfaction with medical staff**							0.02
Very unsatisfied	1	0.3	2	1.3	3	0.6	
Unsatisfied	2	0.6	2	1.3	4	0.8	
Neutral	6	1.7	10	6.6	16	3.2	
Satisfied	78	22.4	26	17.2	104	20.8	
Very satisfied	261	75.0	111	73.5	372	74.6	
**Difficulty visiting health facilities**							0.42
Extremely difficult	1	0.3	2	1.3	3	0.6	
Highly difficult	3	0.9	3	2.0	6	1.2	
Moderately difficult	14	4.0	4	2.7	18	3.6	
Slightly difficult	49	14.1	18	12.1	67	13.5	
Not difficult	280	80.7	122	81.9	402	81.1	
**Difficulty receiving results at health facilities**							
Extremely difficult	137	40.2	58	38.9	196	39.8	0.65
Highly difficult	3	0.9	3	2.0	6	1.2	
Moderately difficult	12	3.5	5	3.4	17	3.5	
Slightly difficult	36	10.6	11	7.4	47	9.6	
Not difficult	153	44.9	72	48.3	225	45.9	
	**Mean**	**SD**	**Mean**	**SD**	**Mean**	**SD**	***p*-Value**
**Number of services used in last 12 months**							
Inpatient services	0.3	1.0	0.4	1.3	0.3	1.1	0.43
Outpatient services	1.6	2.8	1.4	2.6	1.6	2.7	0.04
Health check-up services	3.5	4.4	4.1	4.6	3.6	4.4	0.13

**Table 4 ijerph-16-00069-t004:** Factors associated with HRQOL.

Characteristics	EQ-5D Index ^a^		EQ-VAS Score ^b^	
	Coefficient	95% CI	Coefficient	95% CI
**Sex**				
Female	ref		ref	
Male	0.17 ***	0.09, 0.25	4.69 *	−0.17, 9.54
**Age**	−0.01 ***	−0.01, 0.00	−0.29 **	−0.53, −0.05
**Religion**				
No	ref		ref	
Yes	0.06 *	−0.01, 0.13	−13.33 ***	−18.74, −7.91
**Occupation**				
Retired	ref		ref	
Self-employed	−0.11 **	−0.20, −0.02	−7.49 *	−15.03, 0.04
Others			5.09	−1.66, 11.84
**Having caregivers**				
No			ref	
Yes			−9.86 *	−19.95, 0.24
**Poor status**				
Poor				
Non-poor	0.05 *	−0.01, 0.10	10.59 ***	5.98, 15.21
**History of illness**				
Stroke				
No	ref			
Yes	−0.13	−0.30, 0.04		
Anemic cerebral ischemia (Yes vs. No)				
No			ref	
Yes			−5.90	−14.01, 2.21
Diabetes (Yes vs. No)				
No	ref			
Yes	−0.10	−0.23, 0.02		
Chronic lung disease (Yes vs. No)				
No	ref		ref	
Yes	−0.09 **	−0.18, −0.00	−8.26 **	−16.35, −0.16
Arthritis (Yes vs. No)				
No	ref		ref	
Yes	−0.12 ***	−0.17, −0.07	−7.02 ***	−11.15, −2.89
**Alcohol consumption in the last 5 years**				
No drinking	ref			
Former drinking	−0.18 **	−0.33, −0.02	−13.34 **	−24.63, −2.05
Current drinking	−0.10 **	−0.18, −0.01		
**Cigarette consumption in the last 5 years**				
No smoking	ref			
Former smoker	0.15 **	0.02, 0.27	7.32	−2.09, 16.72
Current smoker	−0.09	−0.20, 0.02	−12.38 ***	−21.34, −3.41

*** *p* < 0.01, ** *p* < 0.05, * *p* < 0.1. ^a^ Positive EQ-5D-5L-Index indicates the presence of difficulty with mobility, difficulty with self-care, difficulty with usual activities, pain/discomfort, and anxiety/depression. ^b^ EQ-VAS score ranges from 0 (worst health state) to 100 (best health state).

**Table 5 ijerph-16-00069-t005:** Factors associated with health service utilization by elderly people.

Characteristics	Inpatient Service		Outpatient Service		Having Regular Health Check-Up	
	OR	95% CI	OR	95% CI	OR	95% CI
**Sex**						
Female	ref		ref		ref	
Male	1.48	0.83, 2.63	0.68	0.43, 1.08	1.27	0.75, 2.16
**Age**	0.97	0.94, 1.01				
**Occupation**						
Retired	ref		ref		ref	
Self-employed					0.39 **	0.16, 0.94
Farmers			1.91 ***	1.27, 2.88		
Others	2.19 *	0.94, 5.11			12.42 **	1.62, 95.57
**Poor status**						
Poor					ref	
Near-poor					0.26 **	0.08, 0.89
Non-poor					0.31 **	0.11, 0.85
**Having caregiver**						
No					ref	
Yes					0.24 *	0.04, 1.31
**Having health insurance (Yes vs. No)**						
No			ref		ref	
Yes			3.52 ***	1.94, 6.41	6.06 ***	3.26, 11.28
**EQ-5D index**	0.24 **	0.07, 0.83				
**EQ-VAS**	0.99 **	0.97, 1.00				
**Smoking Status**						
No smoking			ref			
Current smoker			1.97	0.76, 5.11		

*** *p* < 0.01, ** *p* < 0.05, * *p* < 0.1.

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
