# Peer review of "Gender Differences in Quality of Life and Health Services Utilization among Elderly People in Rural Vietnam"

_ijerph, 2018, doi:10.3390/ijerph16010069_

Reviewer 1 Report

The following are some observations and suggestions. Overall I think this was a good study and may need some adjustments in the conclusions as well as more clarification of certain variables as noted below. 

1. The word 'sex differences' is used in the title and it is clear that the male vs. female genders are what are being referred to. However, in several other areas (39, 108  ) the word 'gender' maybe more appropriate. I would suggest the change even in the title to be consistent. 

2. 117- Excluding certain unstable medical conditions such as a recent stroke, MI etc. are rational,  however, to exclude anyone who has CAD, Cardiomyopathy, inability to ambulate etc. would be limiting the information the study is especially attempting to gather. What other diseases have been excluded? It is not clear what the rationale was to exclude why those who cannot walk independently from the study.

3. 132 - alcohol drinking statuses - consider changing to 'consumption'

4. 138-141: Can this tool be truly extrapolated to the Vietnamese patient/culture/health system? the authors conclude it has been validated and reference just one study that used it. 

5. 164-165: The layers to protect subject's privacy and confidentiality do not appear to be rigorous. How was the data secured? How were patient's de-identified? If more was done, then this should be expanded upon. 

6. Table 1: Only 30% of men made up the total 'n'. This limits the conclusions and inferences drawn from the analysis 

7. Table 1 - Define freelancers

8. Qualify poor, near poor and non-poor

9. A large percentage (78.8%) of the n, identified as non-poor. This would imply they had  financial means which would also skew the analysis as they may have had access to other resources for healthcare and it does not reflect a true status of access and quality to health care afforded by the Vietnamese MOH. 

10. Table 2 - HRQOL: Define the categories -- difficulty with mobility - how is this different to excluding those who cannot ambulate independently? Difficulty with 'usual activities'? Are these ADL's?

11. Religion - for the first time in the paper religion appears as a variable. It is not explained anywhere in the paper as to its significance in the data and its analysis. What role did this play in the choices patients made or did it?

12. 259: This could also be due to less self reporting of anxiety and depression by males. 

Author Response

Response to Reviewer 1 Comments

RE: Revision of the Manuscript “Sex Differences in Quality of Life and Health Services Utilization among Elderly People in a Rural District in Vietnam”

Dear Reviewers, 

We are very pleased to have received your letter in a short period of time. We have addressedthereviewers’ concerns, as enumerated in the following attached file. All authors have agreed on these alterations.We would like to submit the revised manuscript and trust that the manuscript is now acceptable for publication in the International Journal of Environmental Research and Public Health.

Thank you very much for your support and we look forward to hearing from you soon.

Best regards,

Huyen Thanh Thi Vu and co-authors

REVIEWER 1

Comments

Responses

Position

The word 'sex   differences' is used in the title and it is clear that the male vs. female   genders are what are being referred to. However, in several other areas (39,   108) the word 'gender' maybe more appropriate. I would suggest the change   even in the title to be consistent. 

Thank you for pointing   this out. We have changed the title accordingly to “Gender Differences   in Health-related Quality of Life and Health Services Utilization between   Elderly Men and Women in a Rural District in Vietnam” for clarity.

We have also made   changes throughout the manuscript accordingly.

Page 1, Lines 2-4

Throughout manuscript

117- Excluding certain   unstable medical conditions such as a recent stroke, MI etc. are rational, however,   to exclude anyone who has CAD, Cardiomyopathy, inability to ambulate etc.   would be limiting the information the study is especially attempting to   gather. What other diseases have been excluded? It is not clear what the   rationale was to exclude why those who cannot walk independently from   the study.

We excluded people   having these conditions because they were not be able to understand correctly   and answer questionnaire independently. We have changed “inability to walk   independently” to “cognitive impairment” to more accurately describe the   finding.

Page 3, line 118

132 - alcohol drinking statuses   - consider changing to 'consumption'

We have changed   “drinking” to “consumption.”

Page 3, Line 131

138-141: Can this   tool be truly extrapolated to the Vietnamese patient/culture/health   system? The authors conclude it has been validated   and reference just one study that used it. 

We have cited another   paper conducted in Vietnamese general population

Page 3, line 145

164-165: The layers   to protect subject's privacy and confidentiality do not appear to be   rigorous. How was the data secured? How were patient's de-identified? If more   was done, then this should be expanded upon. 

We have revised this   part accordingly.

Page 4, line 171-174

Table 1: Only 30%   of men made up the total 'n'. This limits the conclusions and   inferences drawn from the analysis 

We have acknowledged in   the limitations

Page 13, line 334-336

Table 1 - Define   freelancers

We have changed to   “Self-employed”

Table 1

Qualify poor, near poor   and non-poor

It is based on the   Government’s classification. We have noted in the footnote of Table 1.

Table 1

A large percentage   (78.8%) of the n, identified as non-poor. This would   imply they had financial means which would also skew the analysis as   they may have had access to other resources for healthcare and it does not   reflect a true status of access and quality to health care afforded by   the Vietnamese MOH. 

In this study, we did   not limit our data on access to public health care facilities, but also   included other types of facilities such as private health facility or   self-medication in evaluating  HRQOL   and health care utilization. However, we have acknowledged this limitation in   the discussion section.

Table 2 - HRQOL: Define   the categories -- difficulty with mobility - how is this different to   excluding those who cannot ambulate independently? Difficulty with 'usual   activities'? Are these ADL's?

These variables are a   part of EQ-5D-5L instrument. We have added more details in this section.

Page 3, line 136-137

Religion - for the first time in   the paper religion appears as a variable. It is not explained anywhere in the   paper as to its significance in the data and its analysis. What role did this   play in the choices patients made or did it?

We have added data about   religion in Table 1. Religion might affect mental health, thus might affect   HRQOL. Therefore, we included this variable in our analysis

Table 1

259: This could also be due to less self reporting of anxiety and   depression by males.

Thank you for making   this great point! We have added this consideration in the manuscript.

Page 10, Lines 270-272

Reviewer 2 Report

There are some concerns that the authors need to address. 

1) Data and participants:

How were the participants recruited to the study? How many of them were women and men? 

Did the authors identify the participants through chart review? Please clarify. 

2) Methods: this is the most problematic section. 

Measurement

The rationale of using both EQ-5D-5L and EQ-VAS to measure health status is not clear. Please clarify. 

The description between line 139 and 140 on page 3 is confusing. The authors did not compare the "no problem" group to the "having problem" group in their study so what did that mean?

Why were questions on health behaviors (i.e., drinking and smoking) used to measure health status? These variables are correlated with health status but not the 'measure' of health status. 

Health services utilization was not correctly defined or measured. The questions used in the study were for medical service satisfaction rather than the utilization. Health services utilization should be measured in frequency and quantity of service use for health prevention, disease cure, and long-term care. 

Analysis:

The authors mentioned that they "identify factors associated with the EQ-5D index and EQ-VAS scores," (on page 4 line 153) but they did not conduct factor analysis (e.g., EFA) to identify the 'factors' associated with the psychometrics. 

The authors also mentioned "the outcome data were censored" on page 4 line 154, however they did not conduct clinical trial analysis or survival analysis so what did that mean?

3) Results:

ORs for EQ-5D-5L and EQ-VAS were missing in Table 5. And, the results on alcohol consumption were missing in that table.

Author Response

REVIEWER 2

Comments

Responses

Position

Data and participants:

-How were the   participants recruited to the study? How many of them were women and   men? 

-Did the authors   identify the participants through chart review? Please clarify. 

We have added more   details in study design

Page 3, line 119-123

Measurement

-The rationale of using   both EQ-5D-5L and EQ-VAS to measure health status is not clear. Please   clarify. 

We have clarified our   rationale in Method section

Page 4, line 147-150

-The description between   line 139 and 140 on page 3 is confusing. The authors did not compare the   "no problem" group to the "having problem" group in their   study so what did that mean?

We compared the   difference in “Having problems” across domains between males and females,   which we have nowclarified in Data analysis

-Why were questions on   health behaviors (i.e., drinking and smoking) used to measure health status?   These variables are correlated with health status but not the 'measure' of   health status. 

We have moved these   variables to “Socioeconomic and behavior information”

-Health services   utilization was not correctly defined or measured. The questions used in the   study were for medical service satisfaction rather than the utilization.   Health services utilization should be measured in frequency and quantity of   service use for health prevention, disease cure, and long-term care. 

In this study, we   measured whether participants used any types of health service in the past 12   months (inpatient, outpatient and regular health check up service). We also   measured the frequency of service use. We have clarified these results in   Table 3

Page 3, line 133-134

-The authors mentioned   that they "identify factors associated with the EQ-5D index and EQ-VAS   scores," (on page 4 line 153) but they did not conduct factor analysis   (e.g., EFA) to identify the 'factors' associated with the   psychometrics. 

We identified the   factors associated with HRQOL by using multivariate regression as mentioned   in the Data analysis section.

-The authors also mentioned   "the outcome data were censored" on page 4 line 154, however they   did not conduct clinical trial analysis or survival analysis so what did that   mean?

The censored data mean   that they had upper and lower thresholds. For example, EQ-5D-5L had an upper   value = 1 and lower value = -0.452, and we cannot observe any values that   above or below this certain magnitude.

Results:

ORs for EQ-5D-5L and   EQ-VAS were missing in Table 5. And, the results on alcohol consumption   were missing in that table.

In this study, we   applied stepwise forward selection strategies to build the reduced regression   models. In Table 5, OR for EQ-5D-5L and EQ-VAS, or other variables, were   missing due to the fact that their p-values were more than 0.2; therefore, we   have to remove them in the final models.

Reviewer 3 Report

The manuscript examined the sex differences in equality of life with various health outcomes and health service utilization among older people in a rural area of Vietnam.

Title: I think it is concerning about the health-related quality of life rather than general quality of life. I suggest the title adds health-related quality of life instead of “quality of life”.

Abstract: Add very brief sentence for the reason why “there may be unique health profiles and needs.” I do not think just large population is concentrated in rural areas, which cannot serve rationales for the expected uniqueness of health profiles.

Need to specify sex ratio skewness, just saying women population is higher than men population in rural, for example.

Introduction: This is very well organized introduction especially regarding the research background.

The authors mentioned a wide range of health outcomes.  It would be good to clearly mention what health outcomes in the present study would be main focus in the last part of the introduction. I understand this paper mainly relies on overall quality of life as described in the methods.

Materials and Methods:  Why is “SocSon” selected for the study? Why were those with severe health problems excluded? Need to include a brief description of the rationale for the selection of research site and exclusion of samples.

I was wondering how their educational backgrounds are. It is because educational background would be associated with health-seeking behavior and health-relevant knowledge, which in turn affect their health outcomes as well as account for possible differences in health outcomes.

Conclusion: The authors did not address any limitations associated with the paper. 

Author Response

Response to Reviewer 3 Comments

RE: Revision of the Manuscript “Sex Differences in Quality of Life and Health Services Utilization among Elderly People in a Rural District in Vietnam”

Dear Reviewers, 

We are very pleased to have received your letter in a short period of time. We have addressedthereviewers’ concerns, as enumerated in the following attached file. All authors have agreed on these alterations.We would like to submit the revised manuscript and trust that the manuscript is now acceptable for publication in the International Journal of Environmental Research and Public Health.

Thank you very much for your support and we look forward to hearing from you soon.

Best regards,

Huyen Thanh Thi Vu and co-authors

REVIEWER 3

Comments

Responses

Position

Title: I think it is   concerning about the health-related quality of life rather than general   quality of life. I suggest the title adds health-related quality of life   instead of “quality of life”.

Thank you for pointing   this out. We have changed the title accordingly to “Gender Differences   in Health-related Quality of Life and Health Services Utilization between   Elderly Men and Women in a Rural District in Vietnam.”

Page 1, Lines 2-4

Abstract: Add very brief   sentence for the reason why “there may be unique health profiles and needs.”   I do not think just large population is concentrated in rural areas, which   cannot serve rationales for the expected uniqueness of health profiles.

Need to specify sex ratio   skewness, just saying women population is higher than men population in   rural, for example.

We agree with your   suggestion and have incorporated it. In addition, we have added a few clauses   in the introduction to further strengthen this rationale.

Page 1, Lines 35-37

Page 2, Lines 79-83

Introduction: This is   very well organized introduction especially regarding the research   background.

The authors mentioned a   wide range of health outcomes.  It would be good to clearly mention what   health outcomes in the present study would be main focus in the last part of   the introduction. I understand this paper mainly relies on overall quality of   life as described in the methods.

Thank you for the   suggestion – we have added in a sentence that to better clarify what sorts of   measures this study examines.

Page 3, Lines 107-109

Materials and   Methods: 

-Why is “SocSon”   selected for the study? Why were those with severe health problems excluded?   Need to include a brief description of the rationale for the selection of research   site and exclusion of samples.

Soc Soc district is a   typical rural area setting in Vietnam; therefore, it is   selected for study setting. We have included the rationale of exluding people   with these health conditions in the sampling section

Page 3, line 117-120

-I was wondering how   their educational backgrounds are. It is because educational background would   be associated with health-seeking behavior and health-relevant knowledge,   which in turn affect their health outcomes as well as account for possible   differences in health outcomes.

Thank you very much.   This information was not collected in this study. We have acknowledged this   limitation in Discussion section.

Page 12, line 336-337

Conclusion: The authors   did not address any limitations associated with the paper. 

We have addressed   limitations in the Conclusions section per reviewer’s suggestion

Page 12, Lines 336-339

Round  2

Reviewer 2 Report

Although the authors addressed some of my comments, some issues are still not explicated. 

Data: what's the size of the sample pool from which the authors randomly selected 530 older participants? 

The measure of health care service utilization is still not correct in the study. How were the health care service utilization measured? Were the variables used to measure the utilization categorical or numerical? Again, the whole set of health care satisfaction variables are factors that would affect health care utilization rather than the direct measures of utilization. The authors should have controlled for the effect of these variables in their regression model. 

"Health status" and "HRQOL" and "sex" and "gender" were used interchangeably. Please be consistent throughout the paper.

The authors said that they clarified the impact of gender on the outcomes in the Data analysis, but there is no description about gender in that section. 

On page 4, line 185-187, the authors said " to identify factors associated with the EQ-5D index and EQ-VAS scores, we used the multivariate regression." In my previous comments, I said this is very confusing because the authors sounded doing the factor analysis but they actually did not. Please either delete it or rephrase the sentence. 

Lines 174-176 on page 4 should be moved to the beginning of the paragraph. 

Editing is needed for this paper. 

Author Response

Dec 16, 2018

Dear Reviewers, 

Thank you for your additional comments! We have addressed these concerns, as enumerated in the following attached file. All authors have agreed on these alterations. We would like to submit the revised manuscript and trust that the manuscript is now acceptable for publication in the International Journal of Environmental Research and Public Health.

Thank you very much for your support and we look forward to hearing from you soon.

Best regards,

Huyen Thanh Thi Vu and co-authors

REVIEWER 2

Comments

Responses

Position

Data: what's the size of the sample pool from   which the authors randomly selected 530 older participants? 

It’s around 33,000 thousand elder people who   met our criteria. We have added in the study design

Line 121

The measure of health care service utilization   is still not correct in the study. How were the health care service   utilization measured? Were the variables used to measure the utilization   categorical or numerical? Again, the whole set of health care satisfaction   variables are factors that would affect health care utilization rather than   the direct measures of utilization. The authors should have controlled for   the effect of these variables in their regression model. 

We have clarified the manner to measure health care service utilization in the manuscript. We have also included satisfaction variables in the initial models but they were exclude after using stepwise strategies. Therefore, we did not include these variables in the final models.

Line 151-154

"Health status" and "HRQOL"   and "sex" and "gender" were used interchangeably. Please   be consistent throughout the paper.

In the Introduction, we use the term “sex”   solely when discussing sex ratio, which is the official term used in the   literature to describe men-to-women ratios historically. Meanwhile, in other   parts of the manuscript, we have changed “sex” to “gender”.We also changed   “health status” to “HRQOL”.We hope this clarifies our usage of these terms.

Throughout paper

The authors said that they clarified the impact   of gender on the outcomes in the Data analysis, but there is no description   about gender in that section. 

Thank   you very much. We have added this description in Data analysis

Line 161-162

On page 4, line 185-187, the authors said   " to identify factors associated with the EQ-5D index and EQ-VAS scores,   we used the multivariate regression." In my previous comments, I said   this is very confusing because the authors sounded doing the factor analysis   but they actually did not. Please either delete it or rephrase the   sentence. 

We have revised our method used here to   “multivariable regression model”. We have also edited this paragraph

Line 159-170

Lines 174-176 on page 4 should be moved to the   beginning of the paragraph. 

We have moved as suggestion.

Line 172-173

Editing is needed for this paper. 

Thanks! We have proofread the paper.